# In Vitro and In Vivo Models to Study Nephropathic Cystinosis

**DOI:** 10.3390/cells11010006

**Published:** 2021-12-21

**Authors:** Pang Yuk Cheung, Patrick T. Harrison, Alan J. Davidson, Jennifer A. Hollywood

**Affiliations:** 1Department of Molecular Medicine and Pathology, The University of Auckland, Auckland 1142, New Zealand; pang.cheung@auckland.ac.nz (P.Y.C.); a.davidson@auckland.ac.nz (A.J.D.); 2Department of Physiology, BioSciences Institute, University College Cork, T12 XF62 Cork, Ireland; p.harrison@ucc.ie

**Keywords:** cystinosis, lysosomal storage disease, cell and animal models

## Abstract

The development over the past 50 years of a variety of cell lines and animal models has provided valuable tools to understand the pathophysiology of nephropathic cystinosis. Primary cultures from patient biopsies have been instrumental in determining the primary cause of cystine accumulation in the lysosomes. Immortalised cell lines have been established using different gene constructs and have revealed a wealth of knowledge concerning the molecular mechanisms that underlie cystinosis. More recently, the generation of induced pluripotent stem cells, kidney organoids and tubuloids have helped bridge the gap between in vitro and in vivo model systems. The development of genetically modified mice and rats have made it possible to explore the cystinotic phenotype in an in vivo setting. All of these models have helped shape our understanding of cystinosis and have led to the conclusion that cystine accumulation is not the only pathology that needs targeting in this multisystemic disease. This review provides an overview of the in vitro and in vivo models available to study cystinosis, how well they recapitulate the disease phenotype, and their limitations.

## 1. Introduction

Infantile nephropathic cystinosis is a rare, hereditary, autosomal recessive, lysosomal storage disease affecting 1 in 100,000–200,000 live births [1]. It is caused by mutations in the gene *CTNS* which encodes for cystinosin, a cystine-proton cotransporter found on the lysosomal membrane [2,3]. Worldwide, the most common mutation is a homozygous 57 kb deletion that eliminates the first nine exons and part of exon 10 of the gene and part/all of neighbouring genes *TRPV1* and *CARKL,* respectively [4]. Cystinotic patients are usually asymptomatic at birth and develop normally until the first 6 months of life, when they present with failure to thrive, excessive thirst and urination, dehydration, and sometimes rickets. These symptoms result from Fanconi syndrome, which is the excessive urinary loss of electrolytes such as glucose, phosphate, amino acids, bicarbonate, and low molecular weight proteins, as a consequence of renal proximal tubule dysfunction [5,6]. Schneider et al. (1967), were the first to show that cystinosis is characterised by the accumulation of cystine within lysosomes [7], and today the disease is recognised as the most common cause of inherited renal Fanconi syndrome [8]. The Fanconi syndrome in cystinosis is accompanied by the presence of “swan-neck” lesions in the kidney cortex, which are caused by atrophy of the epithelial cells of the proximal tubule. Cystinotic children later develop glomerular dysfunction with progressive glomerular podocyte injury, as well as multinucleated podocytes. If untreated, cystinosis progresses to end-stage kidney disease (ESKD) by the end of the first decade of life [9,10]. Other non-renal symptoms include photophobia due to cystine crystal deposition in the cornea, hypothyroidism, bone deformities, stunted growth, cognitive impairment, muscle wasting and, in males, infertility due to hypogonadism [5,11,12]. In addition to infantile nephropathic cystinosis, which is the most frequent form of the disease, there are two milder forms: late-onset juvenile nephropathic cystinosis and ocular cystinosis. The former patients are usually diagnosed at an older age and develop photophobia, mild proximal tubule dysfunction, and ESKD at a slower rate compared to infantile nephropathic cystinosis [6]. In the latter, patients display ocular symptoms with no renal phenotype and are generally diagnosed in adulthood [13]. In this review, we will focus on infantile nephropathic cystinosis.

Currently, the only treatment available for cystinosis is cysteamine, an aminothiol that depletes lysosomal cystine by entering the lysosome and generating cysteine and a cysteine-cysteamine mixed disulfide [14]. The cysteine-cysteamine mixed disulfide exits the lysosome by the cationic amino acid transporter, PQLC2, while free cysteine has been found to be freely removed, most likely via a cysteine-specific lysosomal transport system, thus by-passing the need for cystinosin [5,15,16]. Long-term use and compliance of cysteamine has been shown to improve the prognosis of cystinosis with early and continuous treatment delaying progression to ESKD by six to ten years [1,17,18] and improving glomerular function, quality of life and decreasing the incidence of nonrenal complications [18,19,20,21,22]. However, while cysteamine therapy has substantially improved patient outcomes, it does not reverse the Fanconi syndrome, and most patients eventually require a kidney transplant. The suboptimal benefit of cysteamine has been attributed to its low compliance due to severe side-effects such as unpleasant body odour, gastrointestinal problems such as nausea and vomiting, and requirement for large (1.95 g/m^2^/day) and frequent dosing (every 6 h) [23]. A slow-release form of cysteamine was approved by the FDA in 2013 that requires dosing every 12 h, thus improving the quality of life of patients. However, the impact of the delayed released form of the drug on the disease is similar to cysteamine and does not prevent Fanconi syndrome [24,25]. It is clear that cysteamine cannot reverse the renal manifestations, suggesting that cystinosin may have alternative cellular functions beyond its role as a cystine transporter. Indeed, it has been shown that some of the cellular defects observed in cytsinotic cells, such as autophagy and endocytosis, are not rescued with cysteamine treatment [26,27,28,29]. The development of better therapies that also target these alternative pathways necessitates in vitro and in vivo models that closely mimic the human disease. Over the years, several models have been developed, including in vitro systems that use cells from individuals with cystinosis, as well as genetically modified animal models (Figure 1). In this review, we examine the current models available for the study of cystinosis and evaluate how well they recapitulate the human disease. Furthermore, we explore how these models are helping researchers gain a better understanding of the disease and develop improved therapies.

## 2. In Vitro Models

### 2.1. Primary Human Cells

The pioneering studies of the 1960s and 1970s were performed on human cystinotic white blood cells, fibroblasts, or lymph nodes, as these were easily attainable. This early work was pivotal in discovering the basic underlying cause of cystinosis. Whole leukocytes, polymorphonuclear (PMN) leukocytes or lymphocytes obtained from cystinotic patients as well as fibroblasts from skin biopsies showed that cystine levels in these cells were 80–100 times greater than in control subjects [7,30]. In 1968, examination of lymph nodes from cystinotic patients using electron microscopy revealed that cystine accumulation occurred in lysosomes, as seen by the presence of cystine crystals [31]. An early model of cystinosis was the use of cystine dimethylester (CDME) to artificially load lysosomes with cystine, thereby mimicking the primary phenotype of this disease [32]. Using this method, studies on lysosomes from isolated leukocytes demonstrated that cystinosis arises, at least in part, from an absence of cystine exodus from the lysosome as a result of defective transport [33,34,35,36]. However, CDME loading is now largely discredited as a method to model cystinosis as it does not take into account the lack of a functional cystinosin transporter, it exerts direct toxicity on cells, inhibits mitochondrial ATP production and does not accurately mimic the pathophysiology of cystinosis in vivo [37,38].

Further studies conducted in primary fibroblasts showed that the source of the cystine is from the degradation of cystine-rich proteins and not from oxidised cysteine [39]. However, a recent study by Adelmann et al. (2020) showed that, in fact, a substantial amount of cystine derives from oxidised cytosolic cysteine imported into the lysosomes via the MFSD12 transporter [40]. Interestingly, they found that knockdown of MFSD12 resulted in reduced cystine accumulation in cystinotic fibroblasts. Further investigations are needed to determine if this is a potential therapeutic target for future treatments. The use of polymorphonuclear leukocytes and fibroblasts were critical in the testing of possible treatments for cystinosis. Early studies with reducing agents such as ascorbic acid and dithiothreitol showed that these were ineffective at removing excess lysosomal cystine from fibroblasts and were toxic to the cells [41]. In 1976, Thoene et al. showed that cystine accumulation in fibroblasts and PMN leukocytes could be dramatically decreased when cells were treated with the aminothiol, cysteamine [14,42]. The efficacy of cysteamine is still assessed and monitored in patients today by measuring cystine levels in PMN leukocytes. Primary fibroblasts have also been used to study other mechanisms that may contribute to the pathogenesis of cystinosis, such as mitochondrial dysfunction, glutathione (GSH) production and apoptosis [43,44,45]. Despite cystine accumulation, cystine crystals are not present in primary fibroblasts or leukocytes derived from patients. This could be explained by the fact that these cells are relatively fast growing and short-lived; therefore, there is insufficient time for crystal formation.

Cystinosis is a multisystemic disease with accumulation of cystine in lysosomes throughout the body. However, the kidney is the first organ affected and, therefore, renal cell models have been widely utilised. Renal biopsies and cadavers are good sources of renal cells; however, methods to obtain these cells are invasive or not widely available, respectively [46,47]. Alternatively, methods have been developed to harvest exfoliated proximal tubule cells (PTCs) from urine. Cystinotic PTCs, cultured from urine samples, show a ~100-fold increase in cystine levels compared to control PTCs collected from healthy donors [48,49]. These primary cystinotic PTCs were also found to display a decrease in total GSH levels when compared to control PTCs, whereas this difference was not seen in primary fibroblasts [43,45,50]. The reason for this discrepancy is unknown but may be caused by metabolic differences between each cell type [51].

Investigations by Sansanwal et al. using cystinotic proximal tubule cells and kidney biopsies found high levels of the autophagosome marker LC3-II, and the autophagic substrate p62, compared to healthy controls or cells from patients with non-cystinotic renal defects [52,53]. The colocalisation of LC3-II and p62 in cystinotic cells suggested that there is a block in the autophagy pathway, involved in the recycling of intracellular constituents, a finding that has since been confirmed by other groups [26,29].

Overall, the use of primary cells derived from individuals with cystinosis have been a valuable tool. However, primary cells when grown in vitro can be influenced by external factors. They are slow growing and can only be passaged a limited number of times before losing their epithelial phenotype and undergoing senescence [54]. Regarding primary PTCs, there is the potential for contamination by other cell types such as distal tubule cells and fibroblasts, which might complicate the interpretation of findings. Renal cells in culture have also been shown to decrease oxidative metabolism and rely on glycolytic activity, and this may make any extrapolations to the in vivo setting difficult [55]. Finally, there may be variability between samples due to differences in the patient background, such as age, genetics, and environmental factors.

### 2.2. Immortalised Cell Lines

Immortalised cell lines offer many advantages over primary cells as they are homogeneous and easy to maintain indefinitely, thus overcoming the limitations of cellular senescence seen with primary cells. A number of immortalised PTCs have been established, most notably by transfecting cells with human papillomavirus type 16 (HPV 16) E6/E7 and simian virus 40 large T antigen (SV40T) genes. Both HPV 16 E6/E7 and SV40T achieve immortalisation by deregulating cell cycle checkpoints and preventing growth arrest [56,57].

The human kidney-2 (HK-2) cell line, generated by immortalising healthy human PTCs with HPV 16 E6/E7, has been used in numerous studies including cystinosis [58,59,60,61]. HK-2 cells were employed to explore the notion that Cystinosin may have additional functions outside of the lysosome. HK-2 cells were transfected with a construct encoding green fluorescence protein (GFP) fused to the CTNS-LKG isoform, revealing that this variant of Cystinosin was localised to the plasma membrane [62]. Zhang et al. (2019), used gene editing to disrupt *CTNS* in HK-2 cells and found that these cells accumulate cystine and have impaired chaperone-mediated autophagy (CMA) [63], in agreement with similar observations made in cystinotic mouse fibroblasts [26]. In addition, Zhang et al. (2019), showed that *CTNS* knockout HK-2 cells display decreased expression of *LRP2* and *LC3-II* along with defective LAMP2A and Rab11 trafficking and expression, consistent with disruptions in the endolysosomal compartment [63]. These defects could not be rescued by cysteamine treatment but instead by upregulating CMA using small-molecule activators [63]. These observations provided further evidence that defects in autophagy and endolysosomal trafficking play an important role in the cystinotic phenotype.

Urine-derived cystinotic PTCs that were immortalised with HPV 16 E6/E7, were found to have elevated oxidised glutathione, while total GSH, free cysteine and ATP were all normal, leading to the suggestion that increased oxidative stress may contribute to tubular dysfunction in cystinosis [64]. In later studies, Ivanova et al. (2015) showed that cystinotic PTCs exhibit disorganisation of the endolysosomal compartment, with defective LRP2-dependant endocytosis of proteins and delayed processing of ligands. These latter two cellular phenotypes can be partially restored following treatment with cysteamine [59].

With regards to cystine levels, immortalised cystinotic PTCs load cystine (~0.9 nmol/mg protein) within the range seen in SV40T-immortalised cystinotic fibroblasts (~1.75 nmol/mg protein) and transformed cystinotic lymphoblasts (~0.23 nmol/mg protein) [35,64,65]. However, these levels are lower than those reported in primary cystinotic fibroblasts (2.0 to 6.1 nmol/mg protein), primary cystinotic PTCs (3.48 to 13.8 nmol/mg protein) and in situ kidney (16.7 to 101.7 nmol/mg protein; [1,4,45,48,49,66]. One reason for this variability could be due to the high proliferation rate of immortalised cells, which may limit the amount of cystine that can accumulate within the lysosome [5]. Another cause may be the method used to measure cystine. Earlier studies used protein binding assays and radiolabeling which requires radioactivity, limiting its use, while in later years there has been a shift to more accessible high-performance liquid-based chromatography.

To limit the proliferation rate, a temperature sensitive SV40T variant (SV40tsA58U19) was used to generate conditionally immortalised cystinotic proximal tubule epithelial cells (ciPTECs). This SV40T variant permits cells to proliferate only at lower temperatures of 33 °C. At 37 °C, the large T antigen becomes inactive, and proliferation is halted, allowing cells to mature and differentiate [67]. The use of a temperature sensitive SV40T variant was later combined with transfection of human telomerase (hTERT), which prevents the cells from undergoing replicative senescence [68]. The cystine levels of these cystinotic ciPTECs are 37-fold higher than healthy ciPTECs (~5 vs. 0.14 nmol/mg protein, respectively) and 6-fold higher than PTCs immortalised with HPV16 E6/E7 [64]. Rega et al. (2016), used ciPTECs to investigate the role of transcription factor EB (TFEB), a master regulator of lysosomal biogenesis and autophagy genes [27]. By knocking down *CTNS* with short interfering RNAs they found that cystinotic ciPTECs displayed reduced *TFEB* expression and delayed endocytic cargo processing, and these defects could not be restored with cysteamine. However, inducing TFEB activity reduced cystine accumulation and stimulated cargo processing, further supporting the notion that Cystinosin plays roles beyond cystine transport [27].

Despite their ease of use and availability, caution must be taken when interpreting results with immortalised cell lines. The 2-D environment is not equivalent to the in vivo setting, and this can greatly affect cell behaviour and gene expression [69]. Furthermore, immortalisation can have a profound effect on cell biology. For example, HPV oncoproteins interfere with autophagy, potentially as a way to promote viral replication [70]. In the case of HPV16 oncoproteins, these have been found to activate mTOR complex 1 (mTORC1), a negative regulator of autophagy, and impair autophagosome-lysosome fusion [71]. As cystinosis has been found to impair autophagy, care must be taken with cells immortalized with HPV to avoid confounding effects on the autophagy pathway.

### 2.3. Modelling Cystinosis by siRNA Knockdown

Several groups have used siRNA to transiently knock down *CTNS* in human and animal cells. The first reported study achieved a 50% decrease in Cystinosin protein in HK-2 cells, resulting in increased levels of cysteine and cystine, as well as decreases in both GSH and the oxidised form (GSSG), and a mild increase in the redox state of the Cys/CySS-couple [72]. In a subsequent study looking at aspects of mitochondrial dysfunction, Bellomo et al. (2018) found similar results when comparing *CTNS*-null ciPTECs (homozygous for the 57-kb deletion) and HK-2 cells transiently transfected with siRNA [60]. Their findings included the observation that both cells showed altered mitochondrial function. Specifically, they found that *CTNS*-deficient cells display significantly lower levels of mitochondrial cAMP and a reduction in complex I and V activities and mitochondrial membrane potential [60]. These observations are in keeping with the finding that cystinotic human kidneys show abnormal mitochondrial morphology and increased mitophagy [52]. These defects may be related to the interorganelle communication that occurs between mitochondria and the endolysosomal compartment and the quality control role of lysosomes in degrading defective mitochondria [73]. In support of this, Bellomo et al. found that cysteamine could restore most of the mitochondrial functions [60].

Using rabbit renal proximal tubules, Taub and colleagues (2011) used siRNA-mediated knockdown to investigate the effect of Cystinosin deficiency on epithelial transport, with the goal of understanding the cause of the Fanconi syndrome [74]. They found that knocking down *CTNS* resulted in a 50% decrease in ATP levels, consistent with cystinotic cells having compromised mitochondrial function. They investigated the activity of the basolateral Na^+^/K^+^-ATPase, a major consumer of ATP in PTCs and a prerequisite for driving Na^+^-dependent cotransport at the apical membrane but detected no decrease in activity. Instead, they found a major reduction in the surface localization of the phosphate transport (Slc34A1 aka NaPi2a), accounting for the reduced renal phosphate uptake in cystinosis. However, this defect was not generalizable to all PTC apical proteins, as dipeptidyl peptidase IV was unaffected [74].

In a follow-up study [75], Taub et al. (2012) used the same siRNA knockdown approach to investigate a link between ATP levels and apoptosis in cystinotic cells. They found that AMP-activated protein kinase (AMPK), a sensor of cellular energy change that acts to increase ATP, was activated in *CTNS*-knockdown cells, in keeping with the ATP deficit. Their cystinosin-deficient cells showed an increased sensitivity to apoptosis in response to the nephrotoxin cisplatin, and this could be abrogated by inhibiting AMPK. As AMPK activation has been implicated as an inducer of apoptosis, for instance via phosphorylation of the tumor suppressor p53 [76] they concluded that the ATP deficit seen in *CTNS*-knockdown cells may sensitize these cells to toxicant-induced cell death.

Using siRNA-mediated knockdown of Cystinosin in HK-2, Sumayao et al. (2013) found a similar decrease in ATP levels [38]. They also identified an increase in intracellular O_2_^-^ and NO production, although there was no increase in the overall oxidative stress of the cells in response to H_2_O_2_ challenge. Unlike in other studies, they did not detect a perturbation in autophagy, which led to the suggestion that cystine accumulation may not be obligatory for perturbing autophagy. Alternatively, the discrepancy in their findings may be due to their partial knockdown of Cystinosin (40% reduction in protein) and a corresponding low level of cystine accumulation (only 3-fold higher than control cells), which may be insufficient to illicit a complete cystinotic phenotype.

McEvoy et al. (2015) used siRNA to knock down Cystinosin in a rat pancreatic β-cell line and reported a reduction in ATP production, an increase in oxidative stress and an attenuation of nutrient-stimulated insulin secretion [77]. Although the effect of cysteamine treatment on these cells was not examined, it has been reported that early cysteamine therapy can reduce the incidence of diabetes in individuals with cystinosis and delay the age of onset [18,19].

### 2.4. Induced Pluripotent Stem Cells and Kidney Organoids

Induced pluripotent stem cells (iPSCs) are a valuable tool in disease modelling and have been essential in advancing our understanding of the pathophysiology of other lysosomal storage diseases [78]. Reprogramming of somatic cells such as fibroblasts, with ectopic expression of pluripotent genes, *Sox2, Oct4, Klf4* and *c-Myke*, was first described in 2006 [79]. iPSCs have the ability to self-renew and, under proper conditions, can differentiate into any cell type of the body and produce self-organised three-dimensional tissue structures called organoids. [79,80,81]. Kidney organoids resemble mini organs, mimicking the physiology and the basic structures of the tissue of interest such as the nephron, thus bridging the gap between traditional monolayer cell cultures and in vivo animal models for disease modelling [82]. Using gene-editing technologies such as CRISPR-Cas 9, it is possible to generate isogenic pairs of disease-specific and control iPSC lines and organoids, thus allowing a comparison between identical genetic and epigenetic backgrounds with the only difference being the introduced mutation [83]. This is important, as there can be large variations between iPSC lines, even when they are derived from the same donor cell population.

Hollywood et al., (2020), knocked out *CTNS* (*CTNS*-KO) in human iPSCs using CRISPR-Cas9 with these cells showing ~50-fold higher levels of cystine compared to the isogenic control (~2.5 vs. 0.05 nmol/mg protein) [29]. In addition, this group also generated patient-derived cystinotic iPSCs (*CTNS*^-/-^) by reprogramming stromal cells originating from a cystinotic kidney. Kidney organoids made from these cystinotic iPSC lines showed differential loading of cystine (~9 vs. 2.5 nmol/mg protein for cystinotic vs. control, respectively), enlarged lysosomes, increased apoptosis and increased numbers of autophagosomes indicative of a basal autophagy defect, compared to the controls [29]. Unlike prior cell culture studies, no differences were detected in GSH, ATP or mitochondria morphology. This discrepancy may be due to iPSC metabolism, which relies on glycolysis rather than oxidative phosphorylation for their energy needs [84]. Cysteamine treatment was able to mitigate the cystine and enlarged lysosome phenotypes but failed to correct the apoptosis and basal autophagy defects. While the latter observation may indicate a non-cystine related function of Cystinosin for autophagy, we have found that cysteamine suppresses basal autophagy in iPSCs (unpublished observations). Therefore, more experiments are needed to resolve this issue. Regardless, these data raise important considerations for treating cystinosis with cysteamine, as autophagy is critically important in proximal tubule cells, where it protects them from injury and apoptosis [85]. Hollywood et al., went on to show that treating cystinotic iPSCs and kidney organoids with cysteamine in combination with an mTOR inhibitor, everolimus, which activates autophagy, successfully rescued all of the cystinotic phenotypes. This work paves the way to developing a combination therapy for cystinosis that may be therapeutically superior to cysteamine alone.

While kidney organoids are a promising tool for modelling cystinosis and preclinical drug development, there are some limitations of the system. Most protocols generate kidney organoids that are similar in maturation to second trimester human fetal kidneys. In addition, off-target cell types such as neural, glial and muscle progenitor cells can arise in various proportions and are subject to significant batch-to-batch variation [86]. Extended time in culture does not significantly improve maturation but instead, in some cases, causes undesirable changes such as reduced expression of nephron markers and fibrosis [86,87,88]. Transplantation of kidney organoids into mice leads to improved maturation and vascularisation by the host, but this is technically challenging and limited by the overgrowth of off-target cell types [89,90]. As a result of these drawbacks, cystinotic kidney organoids do not model the tubular degeneration seen in cystinosis including the formation of swan-neck lesions, and further optimisation of organoid protocols are needed [29].

### 2.5. Tubuloids

Tubuloids were first described by Schutgens and colleagues (2019) and represent 3-D clusters of cells derived from primary renal epithelial cells originating from kidney tissue or urine samples. They are of lower complexity than organoids derived from iPSCs and appear as cystic, highly polarised epithelial structures [91]. Nevertheless, tubuloids contain differentiated, functional epithelial cells that express markers of the proximal tubule, loop of Henle, distal tubule and collecting duct, but not podocytes [91,92]. Jamalpoor et al. (2021) generated cystinotic tubuloids and identified high levels of alpha-ketoglutarate (αKG) as a contributing cause of the autophagy and proximal tubule defects [93]. They went on to show that bicalutamide, an antiandrogen medication that is primarily used to treat prostate cancer, significantly reduces αKG levels in cystinotic tubuloids. When bicalutamide was combined with cysteamine, it caused a two-fold more potent reduction in the level of cystine than cysteamine alone. This study demonstrated the power of the tubuloid system to model cystinosis, and as a new platform to undertake preclinical drug testing and develop new therapies for cystinosis [93].

Similar to iPSC-derived kidney organoids, there are some limitations to working with tubuloids. Glomerular cells are lacking and cannot be examined, some transporters are expressed at low levels, suggestive of dedifferentiation or loss of function, and there is a need to culture the cells within an extracellular matrix for their assembly, which adds an additional level of technical complexity compared to kidney organoids [91,92].

## 3. In Vivo Models

Unlike in vitro systems, animal models of cystinosis are able to recapitulate many aspects of the human disease with far greater complexity and in the physiological setting of a whole organism. As cystinosis affects multiple organs and progresses in severity over time, in vivo models are critical for gaining greater insights into the pathophysiology of the disease and for testing the efficacy and side-effects of new therapies. The most widely used in vivo models of cystinosis to date have been mice, rats and zebrafish (see Table 1 for a comparison).

### 3.1. Yeast

The first in vivo model of loss of Cystinosin function was the *Saccharomyces cerevisiae* strain *ers1D*. However, the relevance of this to cystinosis was not appreciated until the human *CTNS* gene was identified as a potential homologue. Full length *CTNS*, but not two cystinosis-causing variants (G308R and L338P), was found to complement the yeast mutant’s sensitivity to the antibiotic hygromycin B [102,103,104]. A variant of *CTNS* that lacks the first 121 amino acids, which are located in the lysosomal lumen, was also able to complement *ERS1* deficiency, demonstrating the importance of the 7-transmembrane region for function. However, the yeast mutant does not abnormally load intracellular cystine and shows no differences in growth and survival compared to wild-type cells, although it is sensitive to oxidative stress [105]. Not unexpectedly then, cysteamine does not chemically complement the defect [104]. Thus, there are some limitations in using the yeast mutant in terms of studying cystinosis. Nevertheless, yeast is genetically tractable, and the complementation assay is well-suited to interrogate Cystinosin variants such as I133F and S298N, which cause severe disease but yet retain 80–100% cystine transport activity, or G197R, which has 20% cystine transport activity but only gives rise to the ocular form of cystinosis [106]. The yeast model has also been used to identify MEH1, a potential regulator of ERS1; therefore, it could be a useful tool to identify new interactors that may play a yet uncharacterised role in cystinosis [104].

### 3.2. Mouse

#### 3.2.1. C57BL6/129sv-

The first *Ctns* knockout (*Ctns*^-/-^) mouse model was generated on a mixed C57BL6/129sv background using a “promoter trap” approach that resulted in a truncated nonfunctional protein which failed to localise to the lysosome [94]. *Ctns*^-/-^ mice display a ~40-fold increase in cystine levels in several organs, including the kidney, with loading increasing with age, compared to wild-type littermates [94]. Mild cystine crystal deposition is observed in interstitial cells of various tissues at 6-months of age including focal crystal deposits within proximal tubule cells. At this stage, behavioural anomalies also become apparent, with reduced motility compared to wild-type littermates. Slit lamp examination of 8-month-old *Ctns*^-/-^ mice revealed cystine crystals in the cornea and bone abnormalities including decreased bone density and cortical width were observed. Additional bone deformities involving the tibia and femur are seen in the knockouts at 9-months of age [94]. However, despite accumulating high levels of cystine and the formation of cystine crystals in the kidney, no signs of proximal tubulopathy are detected either histologically or biochemically, and renal failure does not manifest in the knockouts up to 18-months of age [94]. The lack of correspondence between the mouse knockout and the progression and severity seen in the human disease was disappointing and suggested that the mixed C57BL6/129sv background may not be optimal for modelling cystinosis due to modifier genes.

#### 3.2.2. C57BL/6 and FVB/N-

By backcrossing the C57BL6/129sv *Ctns*^-/-^ strain onto the C57BL/6 and FVB/N backgrounds, two new cystinotic mouse models were subsequently developed [95]. Characterisation of these mice confirmed that the genetic background has a profound effect on the cystinotic phenotype [107]. While both C57BL/6 and FVB/N *Ctns*^-/-^ mouse models display cystine loading in various organs, significant differences in phenotype severity exist between the two lines. For instance, on the C57BL/6 background the *Ctns* mutation results in higher kidney cystine levels compared to the FVB/N background, and manifests with “failure to thrive” and renal dysfunction phenotypes that are not seen with the FVB/N line [95]. Thus, the most severe cystinotic phenotype is observed on a pure C57BL/6 background, with much milder pathologies arising on the mixed (C57BL6/129sv) and FVB/N backgrounds.

The kidney abnormalities in C57BL/6 *Ctns*^-/-^ mice are first apparent at 2-months of age with polyuria and the onset of Fanconi syndrome (increased excretion of glucose, phosphate, potassium, and low molecular weight proteins). Starting at 6-months of age, the renal cortex shows a loss of expression of proximal tubule apical transporters such as *LRP2*, reduced tight junction integrity, and “swan neck” lesions. By 9-months of age, the histological defects also include tubular atrophy, denuding of the proximal tubule epithelium and thickening of the tubular basement membranes. Renal function, as measured by urine levels of urea and creatinine, declines steadily from 10–18 months of age, consistent with chronic renal disease progressing to kidney failure [95,96,108].

Despite these morphological changes, the C57BL/6 *Ctns*^-/-^ model does not completely recapitulate the human disease. Notably, the Fanconi syndrome does not include a loss of bicarbonate, amino acids and sodium. In addition, C57BL/6 *Ctns*^-/-^ mice do not show podocytes abnormalities, unlike cystinotic patients where glomerular defects include foot process effacement and multinucleated podocytes [10,95,109]. In addition, C57BL/6 *Ctns*^-/-^ mice develop Fanconi syndrome prior to the histological signs of proximal tubule atrophy whereas in human patients, Fanconi syndrome is concomitant with proximal tubule atrophy and the formation of “swan-neck” lesions [9,96]. This difference in timing may reflect a greater resolution of disease progression in mice and could be exploited to better understand the molecular changes governing the transition from dysfunctional transport activity to tubular atrophy. It should also be noted that subsequent publications by different groups using the C57BL/6 *Ctns*^-/-^ mice have reported varying degrees of Fanconi syndrome, from mild/incomplete to nonexistent [110]. This variation may be due to inbreeding at different facilities.

Despite these differences, this mouse model has been instrumental in expanding our understanding of the pathogenesis of cystinosis [26,96,108,111]. Gaide Chevronnay et al. (2014) showed that cystine loading resulted in early loss of the SLC5A2 (SGLT-2) and SLC34A3 (NaPi-11a) transporters, the brush border receptors LRP2 and Cubilin, and the LT-lectin, suggestive of proximal tubule apical dedifferentiation [96]. As this dedifferentiation occurred before overt histological lesions in the proximal tubule, it suggests that apical dedifferentiation, rather than cellular atrophy, causes the Fanconi syndrome.

Galaretta et al. (2015) used C57BL/6 *Ctns*^-/-^ mice to study the formation of the “swan neck” lesion and suggested it was an adaptive response to oxidative stress [112]. Subsequent in vitro studies using mouse primary proximal tubule cells from C57BL/6 *Ctns*^-/-^ animals provided further evidence to indicate that the source of the oxidative stress are defective mitochondria that have not been removed due to lysosome dysfunction and the block in autophagy [99]. It was also discovered that increased oxidative stress promotes the phosphorylation of the tight junction protein ZO-1, providing an explanation for the disruption in tight junction integrity. Furthermore, phosphorylation of ZO-1 results in this protein no longer being able to sequester the Y-box transcription factor ZONAB, which moves into the nucleus and promotes proximal tubule cell proliferation and dedifferentiation [99]. These observations help establish a model of cystinotic kidney damage in which (1) cystine accumulation causes lysosomal and autophagic dysfunction, (2) the failure to clear defective mitochondria results in increased oxidative stress (sensitizing the cell to apoptosis), and (3) compensatory proliferation and dedifferentiation induces Fanconi syndrome, compromised epithelial integrity and tubular atrophy. A key protein facilitating this process in proximal tubule cells is LRP2, the multiligand receptor that is responsible for the endocytosis of filtered proteins including albumin, which is cystine rich due to several disulphide bonds [110]. Deleting *LRP2* in C57BL/6 *Ctns*^-/-^ mice via a conditional Cre/Lox approach prevents cystine accumulation, crystal deposition, apical dedifferentiation in the kidney, and the subsequent formation of “swan neck” lesions [110].

Co-immunoprecipitation studies by Andrzejewska et al. (2016), using *Ctns*^-/-^ mouse proximal tubule cells immortalised using the SV40 T antigen, revealed that Cystinosin interacts with components of the vascuolar H^+^-ATPase-regulator complex that controls mTORC1 signalling [28]. This raised the possibility that the mTORC pathway, which is critical for cell proliferation and survival may be altered in cystinosis. They also report that the mTORC pathway is downregulated in these cells and was not rescued with cysteamine treatment. Interestingly, an earlier study by Napolitano et al. (2015), using *Ctns*^-/-^ mouse primary fibroblasts showed no mTORC1 downregulation [26]. This discrepancy may be due to the viral antigens used for immortalisation as these can influence mTORC1 levels [113].

The C57BL/6 *Ctns*^-/-^ mouse model has also been used to gather new insights into non-renal pathologies caused by cystinosis such as bone deformities, thyroid dysfunction, ocular abnormalities and muscle atrophy.

A study by Battafarano et al. (2019) showed that C57BL/6 *Ctns*^-/-^ mice display intrinsic bone deformities as early as 1-month without any renal tubulopathy [111]. They found that cystinotic animals at this age show a reduction in trabecular bone volume, bone mineral density, and number and thickness, as well as an impairment in osteoblasts and osteoclasts, compared with wild-type animals. Gaide Chevronnay et al. (2016) used C57BL/6 *Ctns*^-/-^ mice to investigate the early thyroid changes that lead to the hypothyroidism seen in cystinotic children [114]. They found that 9-month-old cystinotic mice recapitulate key features of human cystinosis-associated hypothyroidism, such as chronically increased levels of thyroid stimulating hormone, follicular activation and proliferation, and eventual thyrocyte lysosomal crystals. They also gathered important insights into the underlying mechanisms by linking impaired thyroglobulin production/processing to ER stress and activation of the unfolded protein response [114].

A characterisation of the ocular abnormalities in *Ctns*^-/-^ mice found that cystine accumulates in a spatiotemporal pattern that closely resembles that of cystinotic patients [115]. The highest levels of cystine were observed in the cornea and iris with corneal crystals observed abundantly from 7-months of age (although mild photophobia was noted from 3-months of age). Only rare retinal crystals were detected (at 19-months of age), coinciding with degeneration of the retinal pigmented epithelium. By contrast, in humans this phenotype can be observed as early as infancy and can precede corneal changes [115,116].

Investigations by Cheung et al. (2016) found that the lower total body mass observed in *Ctns*^-/-^ mice when compared to wild-type littermates was due to increased muscle wasting and energy expenditure. They observed a decrease in muscle mass as well as muscle fibre size along with muscle weakness as indicated by the reduced grip strength and rotarod activity in 12-month-old *Ctns*^-/-^ mice. The authors also observed profound adipose tissue browning as well as the upregulation of genes associated with thermogenesis in both muscle and adipose tissues, both of which contribute to an increase in energy expenditure [117]. Furthermore, these mice were 25(OH)D_3_ and 1,25(OH)_2_D_3_ insufficient, and treatment with vitamin D attenuated the adipose tissue browning and muscle wasting [118].

Finally, the C57BL/6 *Ctns*^-/-^ mouse model has been a critical tool to develop and test new gene and cell-based therapies for cystinosis. Hippert et al. (2008) transduced *Ctns*^-/-^ mice with an adenovirus vector that expresses the wild-type human *CTNS* cDNA and showed that long-term gene transfer (4 weeks) led to a reduction in hepatic cystine levels in young mice (2–3-months old) but not in older mice (>5-months) despite equal transduction efficiencies [119]. As older animals have higher cystine levels and more crystal deposits, they may be either refractory to this kind of therapeutic approach or they may need much longer treatment windows to reduce cystine loads. This study highlighted the importance of early intervention in patients.

Another study showed that transplantation of bone marrow cells from wild-type mice (with normal Cystinosin function) into C57BL/6 *Ctns*^-/-^ mice leads to a >50% reduction in cystine levels in the kidney, eye, heart, liver, spleen and brain, 2 months post-transplantation [120]. Furthermore, the transplanted mice displayed improved renal function as seen by a decrease in serum creatinine and urea compared to nontreated *Ctns*^-/-^ mice. Importantly, these benefits were observed for 13 months, indicating that this approach can confer long-term benefits [120,121]. The effectiveness of this cell-based therapy was found to be dependent on the level of wild-type bone marrow engraftment, with improved kidney function only observed in *Ctns*^-/-^ mice that had >50% donor engraftment [121]. In a follow-up study, hematopoietic stem and progenitor cells from C57BL/6 *Ctns*^-/-^ mice were genetically modified ex vivo to express functional *Ctns* via a lentiviral vector and then transplanted back into knockout animals. This led to abundant integration of transduced blood cells in a range of tissues and, in the highly engrafted animals, this was associated with a reduction in cystine content and improved kidney function compared to nontreated *Ctns*^-/-^ mice [122]. It was subsequently shown that macrophages derived from the transduced blood stem and progenitor cells can generate intercellular bridges called tunnelling nanotubes. These structures are able to transfer lysosomes with functional Cystinosin from the donor cells into the *Ctns*-deficient proximal tubule and thyroid cells, thus explaining the mechanism of rescue [114,123]. As of a result of these ground-breaking, proof-of-principle studies, this gene and cell-based therapy approach has begun phase 1/2 clinical testing (NCT03897361) in individuals with cystinosis.

#### 3.2.3. C57BL/6 Ctns^Y226X/Y226X^ Nonsense Mutant Mouse

A nonsense mutation that creates a premature stop codon in exon 7 of *CTNS* (W138X) is commonly found in cystinosis patients in Canada, making this variant a key therapeutic target for this population. To facilitate this, Brasell et al. (2019) used zinc finger nucleases to generate the equivalent mouse model (*Ctns^Y226X/Y226X^*) on the C57BL/6 background [124]. These animals show a significant reduction in *Ctns* transcript levels and an eight-fold increase in cystine levels in the kidney compared to controls. Prior to 6-months the mutant mice appeared normal, with proteinuria detected at 9-months of age. The renal phenotype was very mild with only male mice showing occasional atrophy of the cells in the Bowman’s capsule. Despite the partial phenotype of these animals, they proved useful to test a novel therapy involving an aminoglycoside (ELX-02) that promotes translational read-through of the stop codon. *Ctns^Y226X/Y226X^* mice treated with ELX-02 were found to have a 30% reduction in kidney cystine levels compared to untreated controls, indicating the therapeutic potential of this approach [124].

### 3.3. Zebrafish

The zebrafish is a valuable model for modelling human diseases as it is inexpensive to house, undergoes rapid organogenesis and the pronephric kidney of the larval zebrafish has some structural and functional similarities to the mammalian nephron [125]. Two zebrafish *ctns* mutants have been generated, both displaying high levels of cystine at the larval stage compared to wild-type controls and responding to cysteamine added to the swimming water [97,99]. Other characteristics of *ctns^-/-^* larvae (seen in either one or both mutant lines) that agree with mammalian models of cystinosis include increased rate of apoptosis in proximal tubule cells, reduced LRP2 (and a corresponding impairment in proximal tubule cell endocytosis), enlarged lysosomes, increased autophagosomes in the proximal tubule cells and partial podocyte foot process effacement. This latter phenotype was associated with a leaky glomerular filtration barrier. Given that podocytopathy is not observed in C57BL/6 *Ctns*^-/-^ mice, even in advanced age, the zebrafish model may be particularly suited to studying the effect of cystinosis on podocytes [97].

Follow-on analysis of adult *ctns^-/-^* zebrafish found that the highest level of cystine accumulation was in the kidney [98]. This result differs from C57BL/6 *Ctns*^-/-^ mice where greatest cystine loading is seen in the liver and spleen. However, the zebrafish mesonephric kidney is also the site of hematopoiesis, which may explain the species difference [95]. Surprisingly, adult *ctns^-/-^* zebrafish do not show podocyte foot process effacement and only a mild enlargement of Bowman’s capsule and the glomerular tuft [98]. This difference compared to the larval mutants may be related to differences in glomerular filtration rates, as larval zebrafish have a single glomerulus that may operate under greater stress compared to adult glomeruli. Adult *ctns^-/-^* zebrafish displayed some additional phenotypes that were not seen at the larval stage, which included ocular anomalies (increased thickness of the corneal stromal layer), decreased fertility and decreased locomotor activity. Unexpectedly, unlike human individuals with cystinosis and the C57BL/6 *Ctns*^-/-^ mouse, the zebrafish mutant showed increased body weight and length compared to healthy controls.

One limitation of the zebrafish *ctns* mutant is a lack of “swan neck” lesions and widespread cystine crystal deposition, although cytoplasmic vacuoles with rectangular polymorphous shapes suggestive of cystine crystals are observed in the proximal tubule cells of adult mutants [98]. It is worth noting that zebrafish, unlike mammals, are able to undergo considerable renal regeneration following injury, and grow new nephrons throughout adult life [126]. Thus, it may not be possible for the full extent of the cystinotic renal pathology to be manifested in this model. Another limitation of the zebrafish in terms of preclinical drug development is their aquatic habit, which make it challenging to deliver compounds and perform pharmacokinetic analyses.

### 3.4. Rat

#### 3.4.1. LEA/Tohm-

The Long-Evans Agouti (LEA/Tohm) rat spontaneous develops renal glucosuria and is used as a model of type 2 diabetes [127]. Using genetic linkage analysis, the causative locus was mapped to a region containing the *Ctns* gene with subsequent sequencing revealing a 13-bp deletion in exon 7 that truncates the protein [100]. The mutation was then bred onto the F344 (F344-*Ctns^ugl^*) and characterised. At 10 months of age, abnormally high levels of cystine were present in several organs including the kidney (4.5-fold over controls). Histopathological signs of proximal tubular atrophy and cystine crystal deposition were found at one year of age, although the precise location of the crystals is unclear. A major drawback of this model is that the phenotype has not been comprehensively characterised. Renal function, LRP2 levels, apoptosis, oxidative stress, and autophagy have not been evaluated, therefore the degree of renal dysfunction and Fanconi syndrome is unknown. In addition, the onset of the disease and its progression were not reported, making it difficult to compare this model to the human disease.

#### 3.4.2. Sprague Dawley

Hollywood et al., generated a new *Ctns* knockout rat model that closely recapitulates the human disease phenotype. Using CRISPR-Cas9 gene editing, frameshift indel mutations were introduced into exon 3 of the *Ctns* gene in the Sprague Dawley (SD) background, resulting in a truncated and non-functional Cystinosin protein [101]. These *Ctns*^-/-^ rats show cystine loading and lysosomal crystal formation in various organs from 3-months of age. Fanconi syndrome manifests from 3–6-months of age as indicated by polyuria and polydipsia and increased urinary excretion of total protein, glucose, albumin and calcium. Histologically, *Ctns*^-/-^ rats display kidney lesions, such as proximal tubule atrophy, basement membrane thickening, and the presence of “swan neck” lesions from 3-months of age, likely coinciding with the onset of Fanconi syndrome. Loss of LRP2 is observed starting in the superficial kidney cortex from 6-months of age, with a progressive decline towards the medulla until LRP2 is only found in the juxtamedullary tubules by 17-months of age [101]. This loss appears to be proceeded by injury, as determined by the upregulation of the proximal tubule injury marker, Harvcr1, in tubules lacking LRP2 [101]. Increased urinary levels of phosphate, as well as the decreased urea and creatinine excretion, are detected from 9-months of age onwards, together with increased levels of plasma creatinine, signifying impaired renal function. At this stage, *Ctns*^-/-^ rats also show glomerular lesions, such as multinucleated podocytes and podocyte effacement, which are not seen in C57BL/6 *Ctns*^-/-^ mice but are a feature of the human disease [109]. By 12-months of age, most of the glomerular tufts appear shrunken [10,101]. *Ctns*^-/-^ rats display other non-renal pathophenotypes, such as failure to thrive, crystal deposition in the cornea and changes in the bone microstructure including cortical bone cross-sectional thickness, trabecular thickness, and cortical tissue mineral density [101]. Unlike the mouse knockout, behavioural defects are not observed in *Ctns*^-/-^ rats. Overall, the phenotype of this new *Ctns*^-/-^ rat model closely recapitulates the time course and severity of cystinosis in humans, making it a valuable new tool for the field.

### 3.5. Drosophila

CRISPR-Cas9 gene editing was used to create a fruit fly mutant in *CG17119*, the *Drosophila melanogaster* ortholog of *CTNS*, as part of a larger effort to understand the mTOR pathway and the growth requirements for different amino acids [128,129]. *Drosophila* Cystinosin localizes to lysosomes, and mutant larvae accumulate intracellular cystine. Under starvation conditions, cells lacking *Ctns* in the fat body (similar to white adipose tissue and the liver in humans) show increased mTORC1 signaling and decreased autophagy [129]. In addition, fasted mutants display a developmental delay and a shortened life span. This phenotype can be rescued by treatment with cysteamine, linking it to cystine accumulation. However, the mutants can also be rescued with rapamycin, which inhibits mTORC1 and activates autophagy. These observations agree well with the study of Hollywood et al. (2020), where the mTORC1 inhibitor Everolimus was used in combination with cysteamine to completely rescue cystinotic iPSCs [29], thus providing further evidence to indicate the importance of abnormal autophagy in the cystinotic phenotype.

## 4. Conclusions

The development of iPSC, primary and immortalised cell cultures, as well as a wide spectrum of animal models (yeast, *Drosophila*, zebrafish, mouse and rat), have all contributed to our current understanding of the molecular basis of cystinosis. The emerging consensus is that cystine accumulation in lysosomes leads to disruptions in the endocytic, autophagic and mitochondrial compartments, with varying levels of oxidative stress, energy deficit and sensitivity to apoptosis, depending on the tissue. Additional data suggests that cystine accumulation may not be responsible for all of the phenotypes observed in cystinosis. There is still much to be understood about the nontransport functions of Cystinosin, such as its involvement in the mTORC1 pathway, and it is hoped that these non-cystine-related functions will lead to better treatments. It is clear that no one model of cystinosis is perfect and each has its own associated benefits and limitations. The choice of model will depend on the aspect of the disease being studied and how well it is recapitulated relative to the cystinotic phenotype seen in humans. Care needs to be taken with interpreting the findings, as the levels of cystine loading and degree of cellular dysfunction varies between cell types (fibroblasts vs. renal; primary vs. immortalised) and is subject to genetic modifiers (mouse and potentially rat). Ultimately, the use of these models should inform the development of new treatments for cystinosis. Underscoring this need is the fact that despite long-term cystine depleting therapy, the Fanconi syndrome is not prevented or reversed, and kidney function continues to progressively decline in patients with nephropathic cystinosis. The C57BL/6 *Ctns*^-/-^ mouse has been critical in this regard, as it was instrumental for the preclinical testing of the new gene therapy treatment that is in clinical trials. The SD *Ctns*^-/-^ rat, with its phenotype closely mirroring the human disease, and the physiological advantages of rats for pre-clinical testing, should greatly facilitate the development of additional novel treatments for cystinosis.

## Figures and Tables

**Figure 1 cells-11-00006-f001:**
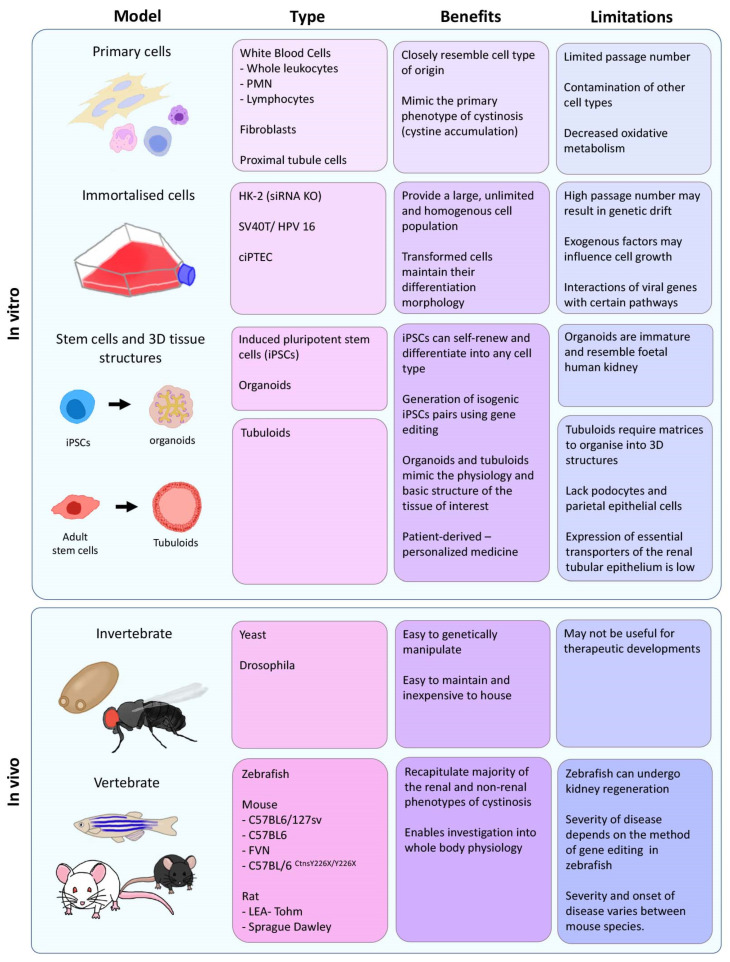
Schematic overview summarising the different in vitro and in vivo cystinotic models. PMN = polymorphonuclear; HK-2 = human kidney-2; SV40T = simian virus 40 large T antigen; HPV 16= human papillomavirus type 16; ciPTEC = conditionally immortalised proximal tubule epithelial.

**Table 1 cells-11-00006-t001:** Comparison of cystinotic in vivo models PTC = proximal tubule cells, dpf = days post fertilisation, N.I = not investigated.

Species	Strain	Mutation in Ctns	Phenotypes								Reference
			Cystine Accumulation	Cystine Crystal	Renal Failure	Glomerulus Changes	PTC Lesions	PTC Dysfunction	Ocular Abnormalities	Bone Deformities	
**Mouse**	C57BL6	IRES- βgal-neo cassette to remove the last 4 exon of Ctns	Yes	Yes	Yes (mild, onset at 10 months of age)	No	Yes (onset at 6 months of age)	Yes (partial, onset at 2 months of age)	Yes	Yes	[94,95,96]
	FVN	IRES- βgal-neo cassette to remove the last 4 exon of Ctns	Yes	Yes (but mild)	No	No	No	No	N.I	N.I	[95]
**Zebrafish**	larvae	homozygous nonsense mutation in exon 8	Yes	No	Yes (decreased inulin clearance)	Yes (podocyte foot effacement)	No	Yes (loss of megalin expression in PTCs)	N.I	N.I	[97]
	Adult	homozygous nonsense mutation in exon 8	Yes	Yes	N.I	Partial	Partial	Suggested	Yes	N.I	[97,98]
	larvae	TALEN-drive 8 bp deletion in exon 3	Yes	N.I	N.I	N.I	No	No	N.I	N.I	[99]
**Rat**	F344	13-bp deletion in exon 7	Yes	Yes (only in kidney cortex)	N.I	N.I	Yes	N.I (glucosuria was detected)	N.I	N.I	[100]
	Sprague Dawley	Indel mutations exon 3	Yes	Yes	Yes	Yes	Yes	Yes	Yes	Yes	[101]

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
