# Peer review of "In Vitro and In Vivo Models to Study Nephropathic Cystinosis"

_cells, 2021, doi:10.3390/cells11010006_

Round 1

Reviewer 1 Report

This a thorough review of non-clinical, bench-based research studies into the pathogenic mechanisms of nephropathic cystinosis. It is comprehensive and well written. However, the findings in tissues other than kidney are only mentioned very briefly. Although most of the previous research, has focused on the impairment of renal function, it would be of interest to analyse non-renal findings. Indeed, some of the animal models mentioned in the manuscript also describe their findings in skeletal muscle, cardiac muscle or cornea. Given that, some of these tissues in animal models, as well as individuals with the disease, behave differently than kidneys (for example skeletal muscle not forming crystals), it would be important to know what is happening there.

Author Response

We thank the reviewer for critical reading of our manuscript and their suggestions for improving the manuscript.

We have expanded our section describing the non-renal phenotypes and it now reads from line 449 to 480,

“The C57BL/6 Ctns-/- mouse model has also been used to gather new insights into non-renal pathologies caused by cystinosis such as bone deformities, thyroid dysfunction, ocular abnormalities and muscle atrophy.

A study by Battafarano et al. showed that C57BL/6 Ctns-/- mice display intrinsic bone deformities as early as 1-month without any renal tubulopathy [107]. They found that cystinotic animals at this age show a reduction in trabecular bone volume, bone mineral density, and number and thickness, as well as an impairment in osteoblasts and osteoclasts, compared with wild-type animals. Gaide Chevronnay et al (2016) used C57BL/6 Ctns-/- mice to investigate the early thyroid changes that lead to the hypothyroidism seen in cystinotic children [111]. They found that 9-month-old cystinotic mice recapitulate key features of human cystinosis-associated hypothyroidism, such as chronically increased levels of thyroid stimulating hormone, follicular activation and proliferation, and eventual thyrocyte lysosomal crystals. They also gathered important insights into the underlying mechanisms by linking impaired thyroglobulin production/processing to ER stress and activation of the unfolded protein response [111].

A characterisation of the ocular abnormalities in Ctns-/- mice found that cystine accumulates in a spatiotemporal pattern that closely resembles that of cystinotic patients [112]. The highest levels of cystine were observed in the cornea and iris with corneal crystals observed abundantly from 7-months of age (although mild photophobia was noted from 3-months of age). Only rare retinal crystals were detected (at 19-months of age), coinciding with degeneration of the retinal pigmented epithelium. By contrast, in humans this phenotype can be observed as early as infancy and can precede corneal changes [112,113].

Investigations by Cheung et al (2016) found that the lower total body mass observed in Ctns-/- mice when compared to wild-type littermates was due to increased muscle wasting and energy expenditure. They observed a decrease in muscle mass as well as muscle fibre size along with muscle weakness as indicated by the reduced grip strength and rotarod activity in 12-month-old Ctns-/- mice. The authors also observed profound adipose tissue browning as well as the upregulation of genes associated with thermogenesis in both muscle and adipose tissues, both of which contribute to an increase in energy expenditure [114]. Furthermore, these mice were 25(OH)D3 and 1,25(OH)2D3 insufficient, and treatment with vitamin D attenuated the adipose tissue browning and muscle wasting [115].

Reviewer 2 Report

The authors presented a broad overview of the models currently available for the study of cystinosis. The manuscript is well written but has some improvements to be made.

  1. Authors stated “free cysteine is removed via an unknown transporter” (row 53) but they should at least mention Pisoni et al. J Cell Biol. 1990 Feb;110(2):327-35. doi: 10.1083/jcb.110.2.327 where it was showed a cysteine-specific lysosomal transport system.
  2. Authors should organize better section “Immortalised cell lines”, they describe HK-2 and PTCs in a discontinuous way, please fix it.
  3. Please carefully revise cystine values reported in rows 176 and 178 and correspondent references. Authors should also discuss differences in measurement methods with respect to discrepancies in cystine levels, “the high proliferation rate of immortalised cells” is not the only cause.
  4. References need to be cleaned up, please use a reference manager (Zotero, RefWorks, Endnote, etc.) there are many duplicate refences e.g., #58 and #74; #1 and #66; #110 and #121…

Author Response

We thank the reviewer for critical reading of our manuscript and their suggestions for improving the manuscript. Please see below the changes we have made to address these concerns.

1. Authors stated “free cysteine is removed via an unknown transporter” (row 53) but they should at least mention Pisoni et al. J Cell Biol. 1990 Feb;110(2):327-35. doi: 10.1083/jcb.110.2.327 where it was showed a cysteine-specific lysosomal transport system.

We have revised this sentence and added the reference to Pisoni et al., (ref #16) and it now reads starting on line 52 “The cysteine-cysteamine mixed disulfide exits the lysosome by the cationic amino acid transporter, PQLC2, while free cysteine has been found to be freely removed most likely via a cysteine-specific lysosomal transport system, thus by-passing the need for cystinosin [5,15,16].  

2. Authors should organize better section “Immortalised cell lines”, they describe HK-2 and PTCs in a discontinuous way, please fix it.

We have re-organised this section and it now begins with the description of HK-2 cells followed by cystinotic PTCs from line 212-519

3. Please carefully revise cystine values reported in rows 176 and 178 and correspondent references. Authors should also discuss differences in measurement methods with respect to discrepancies in cystine levels, “the high proliferation rate of immortalised cells” is not the only cause.

We have revised these values and added a more detailed description of the differences. The manuscript now reads from line 239:

“With regards to cystine levels, immortalised cystinotic PTCs load cystine (~0.9 nmol/mg protein) within the range seen in SV40T-immortalised cystinotic fibroblasts (~1.75 nmol/mg protein) and transformed cystinotic lymphoblasts (~0.23 nmol/ mg protein) [35,63,64]. However, these levels are lower than those reported in primary cystinotic fibroblasts (2.0 to 6.1 nmol/mg protein), primary cystinotic PTCs (3.48 to 13.8 nmol/mg protein) and in situ kidney (16.7 to 101.7 nmol/mg protein; [1,4,44,47,48,65]”.

We have also added a sentence to show that the method of cystine measurements may influence the values reported. Line 246 reads;

“Another cause may be the method used to measure cystine. Earlier studies used protein binding assays and radiolabeling while in later years there has been a shift to more accurate high-performance liquid-based chromatography”.

4. References need to be cleaned up, please use a reference manager (Zotero, RefWorks, Endnote, etc.) there are many duplicate refences e.g., #58 and #74; #1 and #66; #110 and #121…

We have corrected references throughout the manuscript and removed any duplicates.

Round 2

Reviewer 2 Report

The authors responded adequately to all requests for changes, improving the quality of the manuscript.

Author Response

Thank you